# Saffron: Chemical Composition and Neuroprotective Activity

**DOI:** 10.3390/molecules25235618

**Published:** 2020-11-29

**Authors:** Maria Anna Maggi, Silvia Bisti, Cristiana Picco

**Affiliations:** 1Hortus Novus srl, via Campo Sportivo 2, 67050 Canistro, Italy; 2Department of Phyisical and Chemical Sciences, University of L’Aquila, Via Vetoio, 67100 Coppito, Italy; 3Department of Biotecnology and Applied Clinical Sciences, DISCAB, University of L’Aquila, Via Vetoio, 67100 Coppito, Italy; s.bisti@team.it; 4National Institute of Biostructure and Biosystem (INBB), V. le Medaglie D’Oro 305, 00136 Roma, Italy; cristiana.picco@ge.ibf.cnr.it; 5Institute of Biophysics, National Research Council, Via De Marini 6, 16149 Genova, Italy

**Keywords:** saffron, crocins, neuroprotective activity, P2X7 receptor, fraction

## Abstract

*Crocus sativus* L. belongs to the Iridaceae family and it is commonly known as saffron. The different cultures together with the geoclimatic characteristics of the territory determine a different chemical composition that characterizes the final product. This is why a complete knowledge of this product is fundamental, from which more than 150 chemical compounds have been extracted from, but only about one third of them have been identified. The chemical composition of saffron has been studied in relation to its efficacy in coping with neurodegenerative retinal diseases. Accordingly, experimental results provide evidence of a strict correlation between chemical composition and neuroprotective capacity. We found that saffron’s ability to cope with retinal neurodegeneration is related to: (1) the presence of specific crocins and (2) the contribution of other saffron components. We summarize previous evidence and provide original data showing that results obtained both “in vivo” and “in vitro” lead to the same conclusion.

## 1. Introduction

Saffron is a spice obtained from the dehydrated stigmas of the flower *Crocus sativus Linnaeus*, a member of the family of Iridaceae and probably the result of intensive artificial selection of the *Crocus cartwrightianus*, native in the Island of Crete. Saffron was first grown in Iran, where currently about 90% of the global production comes from [1]. Other producing countries are Spain, Greece, Italy, Morocco, Egypt, Israel, New Zealand, Australia, Pakistan, and India.

The production process of the spice follows a complex procedure that is articulated in several phases: (a) flower collection, (b) separation of the stigmas or cleaning, and (c) drying and conservation. Each of these steps in the production process of saffron is strongly influenced by the traditions present in the area of cultivation while following general guidelines. The different cultures together with the geoclimatic characteristics of the territory determine a different chemical composition that characterizes the final product, making it distinguishable from others. In addition, changes in the preparation procedures might strongly modify the final composition of chemical components. Saffron is one of the most expensive spices in the world, but high costs lead to a high rate of counterfeiting. The scientific community’s interest in this product, however, is not limited to guaranteeing its authenticity to the consumer. Advanced pharmacological studies have in fact highlighted its numerous beneficial health effects, including a neuroprotective activity on retinal photoreceptors undergoing/exposed to oxidative stress [2]. Multiple ways of actions have been suggested and widely exploited in microarray experiments [3] and in cellular [4] and animal models [2,5,6,7,8]. In addition. interesting data have been obtained in clinical trials with patients affected by either age-related macular degeneration (AMD) or Stargardt [9] and the results are very promising [9,10,11,12].

The chemistry of saffron is complex; this spice has primary metabolites, which are ubiquitous in nature, such as carbohydrates, minerals, fats, vitamins, amino acids, and proteins. A large number of compounds belong to different classes of secondary metabolites, products of metabolism not ubiquitous but important for the development or reproduction of the organism, such as carotenoids, monoterpenes, and flavonoids, including mainly anthocyanins [13].

Carotenoids are the most important constituents of the spice, from which it derives its color. They include fat-soluble ones, such as α- and β-carotene, lycopene, and zeaxanthin, and water-soluble ones like the apocarotenoid crocetin (C_20_H_24_O_4_) and crocins, the polyene esters of the mono- and di-glycoside crocetin.

Crocins are a family of carotenoids unusually soluble in water as they are mono- and di-glycosylated esters of the dicarboxylic acid crocetin [14]. They make up 3.5% of the weight of the stigmas in the plant. The glycosidic carotenoids of saffron, like all glycosides, are usually thermally labile and photochemically sensitive, especially in solution. Like their precursor, crocins exist in the two isomeric forms *13-cis* and *all-trans*. There is a great variety of crocins because there are different combinations of carbohydrates that can go to esterify one or both carboxyl groups and both isomeric forms. Although these crocins differ in substituents and in configuration, they are very similar in their chemical-physical properties and in particular polarity. These similarities make their separation and subsequent identification extremely difficult [15,16,17,18].

Among the oxidation products of carotenoids, we find two compounds: picrocrocin (monoterpene glycoside) and safranal (cyclic monoterpene aldehyde) that give/are responsible for the spice’s bitterness strength and aromatic strength, respectively. According to the most accredited hypothesis, the precursor is considered zeaxanthin, which is broken at both ends by the enzyme CsZCD (*Crocus sativus* zeaxanthin cleavage dioxygenase) to generate the crocetindialdehyde [19], which can be oxidized and esterified by different glucosyltransferases to give the crocins [20], and picrocrocin. Picrocrocin (C_16_H_26_O_7_), which constitutes 3.7% of the weight of the stigma, has been identified only in the genus *Crocus*, of which the only edible spice is *Crocus sativus* L.; therefore, it constitutes the molecular marker of saffron. During the drying process, the β-glucosidase enzyme acts on picrocrocin to release 4-hydroxy-2,6,6-trimethyl-1-cyclohexene-1-carboxyaldehyde (HTCC, C_10_H_16_O_2_) [21]. For dehydration it is transformed into safranal (C_10_H_14_O). This is present with a percentage of 0.02% in the stigma and is the major component of the volatile fraction of saffron.

The main aim of this paper was to provide evidence of the relationship between the chemical composition of saffron and its neuroprotective activity. Here, we used a consolidated animal model of retinal degeneration to test saffron differing in its chemical components to check whether different saffron preparations might have different efficacy. Experiments involving HPLC analysis and animal treatment were performed in parallel. In addition, we wonder whether all the chemical components of saffron are important in supporting its neuroprotective activities. To test this point we separated two fractions to test crocins and other components separately in both cellular and animal models.

## 2. Results

### 2.1. Correlation between Saffron Chemical Composition and Neuroprotective Activity

The saffron samples were analyzed with the chromatographic method described in the previous section. A qualitative identification of the HPLC-DAD chromatographic peaks was performed using literature data, for similar experimental conditions, on the basis of the well-known absorption spectra of the main constituents of saffron, as well as the relative intensities of the peaks and elution order in chromatograms [22,23,24]. Crocins have characteristic UV-vis spectra, both *trans* and *cis* crocins have a very intense absorption band between 400 and 500 nm, and a further band between 260 and 274 nm, but only *cis* crocins have a relative absorption maximum at 326–327 nm [14,22,24,25,26,27]. Since the analytical standards of these molecules are lacking, a method based on the combination of the areas of the HPLC-DAD peaks observed at 440 nm with the coefficients determined by spectrophotometric analysis was used [28] for the quantitative analysis of crocins. The formula used to determine the concentration of crocins is:(1)$$c\;(\mathrm{mg}/g)=\frac{Mw_{i}.E_{1\;\mathrm{cm}}^{1\%}\;\left(440\;\mathrm{nm}\right).A_{i}}{\varepsilon_{t,c}},$$
where *Mw_i_* and *A_i_* are the molecular weight and the percentage peak area, respectively, *E^1%^_1_*
_cm_ (440 nm) is the coloring strength of the saffron sample, and *ε_t,c_* is the extinction coefficient (89,000 M^−1^cm^−1^ for *trans*-crocins and 63,350 M^−1^cm^−1^ for *cis*-crocins).

The results of the experiments carried out by administering different saffron with different contents of crocins to an animal model of retinal-induced degeneration are shown below (Figure 1). Given the high number of tested saffron samples, we concentrated on and quantified the two most abundant crocins: *trans*-crocetin bis (β-d-gentiobiosyl) ester (T1) and *trans*-crocetin (β-d-gentiobiosyl) (β-d-glucosyl) ester (T2).

In Figure 1 and Table 1, the comparison between five experimental groups of rats (five animals per group) is shown. In three groups (saffron 1, saffron 2, saffron 3), the degeneration was induced with intense light damage (LD) and saffron with different crocin contents was administered through the diet:Group saffron 1: Rats treated with saffron having a content of T1 equal to 13% (mg/g) and T2 equal to 5% (mg/g).Group saffron 2: Rats treated with saffron having a content of T1 equal to 14% and T2 equal to 5%.Group saffron 3: Rats treated with saffron having a content of T1 equal to 17% and T2 equal to 8%.Group LD: Rats untreated but subjected to light damage for 24 h (diseased retina).Group control: Healthy animals (healthy retina).

These results show that the neuroprotective activity of saffron depends on the chemical composition of this spice. Looking at Figure 2, it is evident that the ONL of a retina of an animal treated with saffron 3 (neuroprotective saffron) is close in thickness to that of a healthy animal. On the contrary, saffron 1 and 2 (non neuroprotective saffron) do not have neuroprotective activity. The ONL of the two experimental groups 1 and 2 is similar to that of the retina of an animal exposed to LD and untreated. It has to be noted that according to the International Organization for Standardization (ISO) criteria [29] (looking at the coloring strength), all three saffron belong to class I. These results have allowed the filing of an international patent.

### 2.2. Saffron Components on Cellular Models

To evaluate whether the efficacy of saffron treatment is due to the entire chemical composition of spice or mainly due to specific molecular components [30,31,32], we separated a saffron extract into two fractions: one containing the most polar active components (kaempferol derivates and picrocrocin) and another containing the crocins, the most apolar molecules. The two fractions were previously tested on an animal model [6].

Here, we report new data on two different cellular models: the photoreceptor-derived mouse 661W cells and the HEK293 cells permanently expressing P2X7R. Recently, we found a novel mechanism of saffron neuroprotection by acting directly on the ionotropic P2X7 receptor (P2X7R) [4]. In Corso et al. (2016) [4], we showed that saffron protects photoreceptors from ATP-induced cytotoxicity.

First, we tested both fractions on photoreceptor-derived mouse 661W cells. These cells are a good model for in vitro experiments on retina since they show the biochemical and cellular properties of retinal photoreceptors and activate the same apoptotic program in response to different stresses. Moreover, they express P2X7R [4]. Figure 3 shows viability measurements on 661W cells stressed with different concentrations of ATP. Cells were incubated with ATP (5 and 10 mM), saffron, crocins, and picocrocins/kaempferol derivates at concentrations of 25 µg/mL for 24 h. As previously observed for saffron [4], both components alone also did not change the cell viability (98%). The stress induced by ATP was concentration dependent, reducing cell viability from 57% to 20% at 5 and 10 mM ATP, respectively. Both fractions, crocins (Cr) and picocrocins/kaempferol derivates (PC/Canf), were able to protect against ATP stress but in a different manner. In particular, Cr was more effective, increasing the viability to 69% and 27%, respectively, at 5 and 10 mM ATP. A smaller but significant protection was also observed in the presence of PC/Canf, which raised the viability to 67% and 23% at 5 and 10 mM ATP. However, both fractions were less effective than the entire saffron extract (77% and 30%). When cells were incubated with ATP and Cr and PC/Canf together, the viability increased compared to the single fractions but less than with saffron alone.

Similar experiments were conducted in HEK293 cells permanently expressing P2X7R to test the effect of the two fractions on the isolated receptor. Viability measurements were obtained from MTT tests in HEK-P2X7R cells incubated for 24 h with saffron or Cr or PC/Canf and the selective agonist 2′(3′)-O-(4-Benzoylbenzoyl)adenosine 5′-triphosphate triethylammonium salt (BzATP). We used 10 µM BzATP to reduce cell viability almost to 60%. As observed for 661W cells, Cr reduced cell mortality more than PC/Canf but less than saffron (Figure 3B).

The P2X7 receptor is characterized by two states of permeability [33]. A common feature of both conductance states is the elevation of free [Ca^2+^]_i_, which can reach dramatic levels upon repeated or prolonged application of an agonist. As previously shown, micromolar concentrations of the selective agonist BzATP induced an intracellular calcium elevation in HEK293 cells transfected with the full-length rat P2X7R [4]. Here, we tested the effect of the two fractions in HEK-P2X7R cells loaded with FURA2-AM (Figure 4). First, we verified that both Cr and PC/Canf alone did not change intracellular calcium (Figure 4A, inserts in lower panels). On the contrary, when cells were exposed to 3 µM BzATP, the typical biphasic calcium response with a rapid and a slow [Ca^2+^]_i_ rise was observed (Figure 4A, upper left). Finally, cells were exposed to the same concentration of BzATP in the presence of 25 µg/mL Cr and subsequently in the presence of 25 µg/mL PC/Canf. As shown, the two components of saffron produced different effects on the [Ca^2+^]_i_ response evoked by BzATP application; crocins reduced the [Ca^2+^]_i_ rise, similar to that observed with saffron while picocrocins did not. Moreover, crocins slowed the kinetics of the calcium response as previously found in the presence of saffron [4] and confirmed in the second panel of Figure 4. Vice versa, when cells were exposed to fraction 2, the rise component of the calcium response induced by BzATP was slightly accelerated.

### 2.3. Metabolites of Saffron in Animal Tissues

An open problem is to understand how saffron metabolites reach the various tissues after oral intake. This paper shows the results of analyzed saffron metabolites in different tissues in animal models and blood and urine in AMD patients. Here, we report the data discussed but not shown in a previous paper [5]. We provide evidence of the presence of saffron metabolites only in degenerating retinas. We used 15 animals treated with saffron and LD and 5 control animals treated with saffron without damage. All animals were sacrificed in the morning under the same conditions, saffron was administered through drinking water, and the daily dose was dissolved in the volume of water consumed in 24 h.

Chromatographic analysis of the collected samples revealed the following: in all plasma samples, we found crocetin, while no traces of saffron metabolites were found in other tissue samples, except for degenerating retinas (in 7 of 15 animals, traces of the two main crocins the trans-crocetin bis (β-d-gentiobiosyl) ester and trans-crocetin(β-d-gentiobiosyl)(β-d-glucosyl) ester). The results are shown in Figure 5. We did not find any metabolite (crocetin and/or crocins) in the retina of healthy animals.

In addition to animal tissue samples, we analyzed the blood and urine of two patients with AMD, who were treated with saffron for over a year and three healthy volunteers, who took saffron for two weeks at the same dose of the patients (20 mg/die) (data not shown). Samples were taken two hours after the intake of the morning saffron pill. The most interesting aspect is that crocetin was found only in the samples of healthy volunteers; on the contrary, nothing was found in the blood and urine samples of the patients.

## 3. Discussion

Saffron is an ancient spice whose beneficial properties have been known for a long time. In the past years, our laboratory has focused the attention on its ability to protect against neurodegeneration, in particular age-related macular degeneration (AMD) and Stargardt. By using different approaches from in vitro to in vivo experiments, also on patients, we found different mechanisms of action of this spice. However, to date, our comprehension is not yet complete. This is probably due to the complexity of the chemical components present in the stigmas of this spice. The composition of the constituents is like a “digital fingerprint” for each saffron sample and provides information on its geographical origin. Is it possible that these different components give different efficacies to protect photoreceptors from stress? To answer this question, we correlated the chemical components of different saffron with their neuroprotective capacity. In particular, we concentrated on the carotenoids, crocins and crocetin, both showing antioxidant properties and which may also suppress the activation of proinflammatory pathways [34,35,36]. We found that the composition of crocins was important for saffron neuroprotection. Only a saffron sample with a particular concentration of *trans*-crocetin bis (β-d-gentiobiosyl) ester (T1) and trans-crocetin (β-d-gentiobiosyl) (β-d-glucosyl) ester (T2) was able to protects retinal neurons from light damage (Figure 1 and Figure 2). It should be noted that the difference between “neuroprotective” saffron and “non neuroprotective” saffron is due to the small percentage in the crocins concentration. Looking at chromatograms, it is not possible to appreciate the difference in the contents of crocins. It should be noted that we were able to determine the relationship between neuroprotective saffron activity and chemical composition only through experiments conducted in parallel between the chemical characterization of saffron used and an animal model. Experiments performed in vitro confirmed these results: the screening of different saffron on two cellular models stressed by ATP indicated that the “neuroprotective” saffron increased cell viability and decreased calcium entrance significantly, while the “non neuroprotective” saffron did not (data not shown).

Given the importance of the crocin composition to obtain a “good efficient saffron”, it is legitimate to ask whether crocins alone are able to reproduce the effects obtained by the whole spice.

Therefore, two fractions, picocrocin/kaempferol derivates and crocins, were isolated from the stigmas of saffron and parallel experiments were carried out on 661W and HEK-P2X7R cells. The first cellular model was derived from retinal tumors of a transgenic mouse line and showed biochemical and cellular properties of photoreceptors, while the P2X7 receptor has been proposed as a potential therapeutic target in Central Nervous System (CNS) diseases [37,38]. In particular, high levels of extracellular ATP in the retina could be the cause of retinal neurodegeneration [39,40,41]. Previously, saffron was found to act directly on this receptor [3]; therefore, we also asked if specific saffron fractions conserved this characteristic. In both cellular models, the presence of crocins was important for protection from stress induced by ATP (Figure 3). The fraction containing picocrocin and kaempferol derivates was still able to increase cell viability but with less significance. However, both fractions appeared less effective than total saffron: even in the presence of the two fractions, cell viability did not reach the same values obtained with saffron. Moreover, FURA2 experiments showed that crocins were able to reduce ATP-evoked calcium entry while the fraction picocrocin and kaempferol derivates did not (Figure 4). These data strongly suggest that crocins, together with other yet unidentified saffron compounds, directly target P2X7 receptors, inhibiting their activity, which may be a primary cause for their protective effects against ATP-induced cell mortality (Figure 3). Similar effects were previously observed in an LD animal model [6], where the ability of saffron and its different chemical components to reduce the neuroinflammatory response in the retina was evaluated by the quantification of the number of microglia cells. As observed from in vitro experiments, saffron in stigmas had a strong neuroprotective activity. The fraction containing crocins showed neuroprotective effects, although with greater variability; the fraction containing kaempferol derivates and picrocrocin reduced the number of cells of the microglia in the retina but with less significance. All these data highlight the importance of crocins in the neuroprotection of the retina; however, they clearly point out that the single component or different components together were not able to reproduce the effect obtained with the whole saffron extract and confirmed the complexity of its way of action. These results are not surprising, since the two fractions, although representative of the chemical composition of saffron, are obtained by specific extraction methods that may miss certain compounds of the spice.

Different studies have examined the toxicity of saffron. From in vivo studies, saffron has very low toxicity for doses of up to 1.5 g per day, while at high doses, it causes illness: daily doses ≥ 5 g can induce intestinal bleeding [42]. Ayatollahi et al. (2013) [43] excluded side effects in 60 healthy volunteers related to a treatment with saffron (doses ranging from 200 to 400 mg) for 7 days. Accordingly, it is possible to affirm with confidence that in our case, the treatment with saffron showed no side effects at least at the dose used in humans, considering that patients were treated with amounts 10–20 times lower (20 mg/die).

Several studies conducted in mice and rats showed that the metabolic fate of crocetin and crocins is very different from the one of other C40 carotenoids [44,45]. The orally administered crocins are not detectable in the plasma of rats [44], and its concentration does not tend to accumulate after repeated doses of oral crocins. As for humans, it seems that crocins are absorbed very quickly compared to other carotenoids and that they are eliminated within 8 hours. [46]. Our data indicated that crocins and crocetin were present in the retina of LD animals while in healthy animals, saffron metabolites were found only in the plasma; no traces of crocins and crocetin were present in the retina of healthy animals (Figure 5). The most likely hypothesis is that crocins may be resynthesized from crocetin. Crocetin might reach the retina only as a result of damage of the blood-brain barrier [5]. These data were confirmed by the analysis of the blood and urine samples of patients with AMD and healthy volunteers. Crocetin was found only in the samples of healthy volunteers and not in patients. From the results of this study, it is reasonable to think that only patients with AMD quickly process saffron metabolites [47]. Further experiments are necessary to exploit this interesting point as this is only a suggestion.

Saffron appears to have high potential for the treatment of neurodegenerative diseases. Altogether, the results obtained both “in vivo” and ”in vitro” support the hypothesis of multiple and integrated ways of action able to cope with neurodegenerative processes. One possibility might be that saffron globally activates tissue resilience, as it has been recently hypothesized [48]. This complex action might be supported by the integrated activity of the entire molecular composition of saffron with a very precise chemical profile.

## 4. Materials and Methods

### 4.1. Chemical Analysis of Saffron

We used an HPLC method and spectrophotometric analysis to analyze saffron stigmas, from different regions (Abruzzo, Tuscany, Sardinia, Umbria and Sicily) and from foreign countries (Morocco, Iran, Greece, India, New Zealand, Tasmania, Egypt, and Spain).

Sample preparation for spectrophotometric analysis was carried out according to the procedure ISO-3632 [29], but saffron and solvent amounts were reduced proportionally. About 50 mg of saffron stigma were gently ground in a mortar. In total, 10 mg of powdered sample were suspended in a 20 mL volumetric flask filled with 18 mL of distilled water; the suspension was kept under magnetic stirring for 1 h in the dark and finally diluted to 20 mL. The spectrophotometric measurement was carried out on a suitable aliquot of aqueous extract after a 10 fold dilution and filtration on a 0.45 μm Whatman Spartan 13/0.2 RC (Whatman, GE Healthcare Life Sciences, Little Chalfont, UK) cellulose filter. The UV-vis spectra were acquired in the 200–700 nm range with a Cary 50 Probe (Agilent Techologies, Santa Clara, CA, USA) spectrophotometer using a 1 cm pathway quartz cuvette and pure water for blank correction. The spectra were recorded with a 1 nm resolution. Chromatographic analysis was done using method II of the paper [49].

### 4.2. Animal Model

The Italian Ministry of Health (authorization number 83/96-A of 29/11/1996) authorized the experiments on animals. In addition, all procedures were in line with the ARVO Statement for the Use of Animals in Ophthalmic and Vision Research and were approved by the local Ethical Committee of University of L’Aquila. We followed the protocol extensively reported in Maccarone et al. (2008) [2] for both light-induced damage treatment and histological analysis.

#### 4.2.1. Light Damage (LD)

Sprague-Dawley adult albino rats (2 months old) born and raised in our colony at 5 lux mean luminance were moved singularly into a cage with cold-white fluorescent lights placed at the top and at the bottom to ensure an iso-luminance environment (1000 lux) inside the cage. The litter was removed from the cage to prevent rats from hiding their eyes from the light. Light exposure started at the beginning of the day phase in the animal house, therefore immediately after the 12 hours of darkness. Animals were consecutively exposed to 1000 lux light for 24 hours, and immediately after, they were placed back into normal cages and returned to normal conditions.

#### 4.2.2. Saffron Treatment

The animals belonging to the “LD + saffron” group were treated with saffron to test neuroprotection. Stigmas of saffron (previously tested for their chemical composition) were dissolved in water to obtain a suspension, and 1 mg/kg was offered daily to rats, for 7 days before the light damage. Saffron treatment uninterruptedly continued during the whole recovery period up to the sacrifice of the animals (7 days after LD).

#### 4.2.3. Immunohistochemistry and Morphological Evaluation by Quantitative Histology

One week after bright light exposure, the eyes were enucleated and fixed in 4% paraformaldehyde for 1 hour, washed in 0.1 M phosphate-buffered saline (PBS, pH 7.4), and cryoprotected by immersion in 15% sucrose overnight. Eyes were embedded in optimum cutting temperature (OCT) compound (TissueTek; Qiagen, Valencia, CA, USA), snap frozen in liquid nitrogen/isopentane, and cryosectioned at 20 μm. Sections were collected on gelatin- and poly-l-lysine-coated slides. Sections were three 5-minute washed in PBS and counterstaining with DNA-specific label, bisbenzamide (Hoechst) 1:10,000, for 1 minute at room temperature (RT) to measure the thickness of the photoreceptor layer. The outer nuclear layer (ONL) thickness was measured starting at the dorsal edge along the vertical meridian crossing the optic nerve. Measurements were reported at 1-mm intervals (each point was the mean of four measurements at 250 μm intervals). In each retina, we measured two sections. Images were taken using a confocal microscope (Nikon, Tokyo, Japan) and fluorescence microscope (Nikon).

### 4.3. Solid-Phase Extraction (SPE)

Saffron extract was separated into two fractions: one containing the most polar active components (kaempferol derivates and picrocrocin) and another containing the crocins, the most apolar molecules.

The separation of the different chemical components of saffron was carefully managed with the following solid-phase extraction (SPE) procedure: saffron sample (40 mg) was suspended in 40 mL of a 50/50 CH_3_OH/H_2_O *v*/*v* mixture, under magnetic stirring for 1 hour in the dark. Subsequently, it was centrifuged at 1000 rpm for 5 minutes and the supernatant was dried in a vacuum distiller (rotavapor: 30 °C, 150 rpm). The dry sample was dissolved in a volume of 20 mL of deionized water. The sample was passed through an appropriate SPE C18 cartridge (ISOLUTE with 1 g of stationary phase). The extraction steps were:Conditioning: 2 × 5 mL Hexane 2 × 5 mL Methanol (alternating).Loading: 15 mL of extract 40 mg recovered in 20 mL H_2_O.Elution A: 2 × 1 mL H_2_O-EtOH (75:25).Elution B: 2 × 1 mL EtOH (100%).

With elution A, we obtained the fraction characterized by the presence of kaempferol derivates and picrocrocin, and with elution B, we obtained the fraction constituted by *trans* and *cis* crocins. The entire extraction procedure was repeated simultaneously at least three times for each kind of tested sorbent material.

The two fractions were characterized by an HPLC system used for saffron analysis. After the HPLC analysis, the fractions were dried with a vacuum dryer to remove ethanol and were dissolved in water for animal treatment [2].

We evaluated the effect of two main saffron components, crocins (fraction 1) and picocrocins/kaempferol derivates (fraction 2), on two cellular models.

### 4.4. Cell Cultures

The mouse retinal photoreceptor-derived 661W cell line (kindly provided by Dr. Muayyad Al-Ubaidi (University of Oklahoma Health Sciences Center, OK, USA)) was cultured in Dulbecco minimum essential medium supplemented with 10% FBS, 10% l-glutamine, 100 units/mL penicillin, and 100 μg/mL streptomycin (Gibco). Human embryonic kidney cell line HEK293 stably transfected with a pcDNA3 plasmid containing the full-length rat P2X7-GFP cDNA was maintained in Dulbecco’s modified Eagle’s medium/NutrientMixture F-12 Ham supplemented with 10% FBS, 5 mg/mL gentamycin, and 200 mM glutamine.

#### 4.4.1. Viability Assay

Confluent cells were seeded in 96-well culture plates at a density of 5 × 10^3^ cells/well. After 24 h, cells were incubated with saffron and two fractions of saffron, crocins and picocrocins/kaempferol derivates at a concentration of 25 µg/mL alone and with different concentrations of ATP or 2′(3′)-O-(4-Benzoylbenzoyl)adenosine 5′-triphosphate triethylammonium salt (BzATP). Cell viability was assessed 24 hours after cell treatment by measuring the reduction of 3-(4,5-dimethylthiazol-2-yl)-2,5diphenyltetrazolium bromide (MTT) (Sigma-Aldrich). The absorption at 570 nm was measured using a FLUOstar Omega micro-plate reader.

#### 4.4.2. Intracellular Calcium Measurements

Intracellular calcium measurements [Ca^2+^]_i_ were performed by using the fluorescent Ca^2+^ indicator fura-2 AM. Cells were loaded with 5 μM fura-2 AM dissolved in extracellular solution with 0.1% of pluronic acid to improve dye uptake, for 45 minutes at 37 °C. The cell coverslip was placed on the stage of an inverted fluorescence microscope Nikon TE200 (Nikon, Tokyo, Japan) equipped with a dual excitation fluorometric calcium imaging system (Hamamatsu, Sunayama-Cho, Japan). Cells were excited at 340 and 380 nm at a sampling rate of 0.5 Hz, and fluorescence emission, measured at 510 nm, was acquired with a digital CCD camera (Hamamatsu C4742-95-12ER). The external standard solution was composed of (in mM) 135 NaCl, 5.4 KCl, 1 CaCl_2_, 5 Hepes, and 10 glucose at pH 7.3. The fluorescence ratio F340/F380 was used to monitor [Ca^2+^]_i_ changes. Monochromator settings, chopper frequency, and data acquisition were controlled by a dedicated software (Aquacosmos/Ratio U7501-01, Hamamatsu).

Data were analyzed using IgorPro (Wavemetrics. Portland, Oregon). Results are presented as mean ± standard error of at least 4 independent experiments. Statistical analysis was performed using student’s *t* test or one-way ANOVA to compare the different data sets. Differences were regarded as statistically significant for * *p* < 0.05 and ** *p* < 0.01.

### 4.5. Tissue Analysis for Saffron Metabolites

We analyzed saffron metabolites in different tissues in both animal models and in blood and urine of AMD patients. We used an animal model with induced photoreceptor degeneration [2], treated with saffron through the diet at a dose of 5 mg/kg for a week. As an experimental control group, we used animals without degeneration treated with saffron. The analyzed samples of tissue were retina, plasma, urine, kidney, and liver. Plasma and urine samples from healthy volunteers and AMD patients who took saffron for over a year were analyzed. The samples were analyzed using the solid-phase extraction procedure (SPE) of Yamauchi et al. (2011) [30]. The different tissues were combined with 2.0 mL of methanol, then centrifuged (3000 rpm, 10 minutes) and subjected to the extraction procedure. The various eluates were analyzed with the following HPLC system: Sinergy 4 μm Fusion-RP column (250 × 150 nm, Phenomenex), photodiode array detector (DAD, 210–500 nm, Waters) [5].

## Figures and Tables

**Figure 1 molecules-25-05618-f001:**
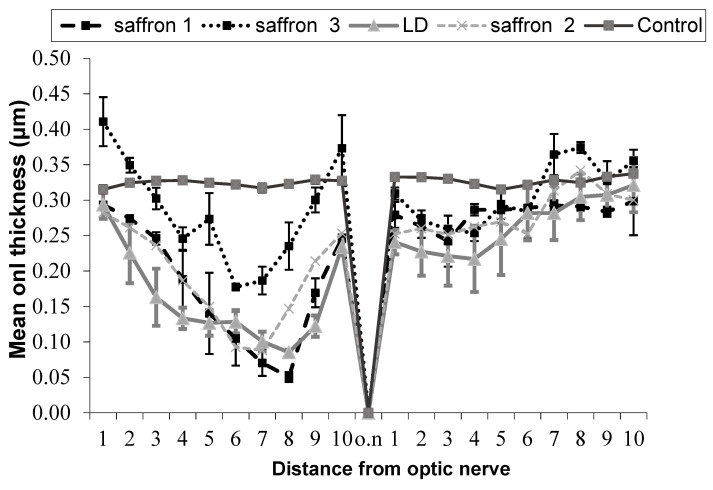
Thickness of the outer nuclear layer (ONL) as a function of the retinal position starting from the dorsal edge of the retina up to the optic nerve entrance and continuing into the ventral retina in 5 experimental groups: healthy animals (control), animals exposed to light damage (LD), and LD animals treated with three different saffron preparations. All animal groups were sacrificed a week after LD. Each point of the graph is the average ±SEM of 5 experiments.

**Figure 2 molecules-25-05618-f002:**
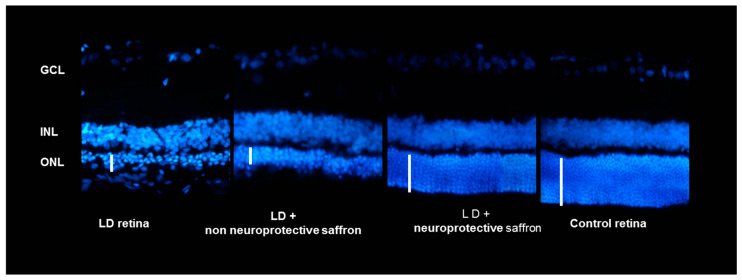
Cross-section of the retina of animals belonging to 4 experimental groups: animals exposed to damage from light not treated with saffron (retinal LD), healthy animals (control retinal), animals exposed to light damage and treated with active saffron (LD+ active saffron), animals exposed to light damage and treated with inactive saffron (LD+ inactive saffron). Images were taken in corresponding dorsal retinal regions. The coloring agent used was bisbenzimide.

**Figure 3 molecules-25-05618-f003:**
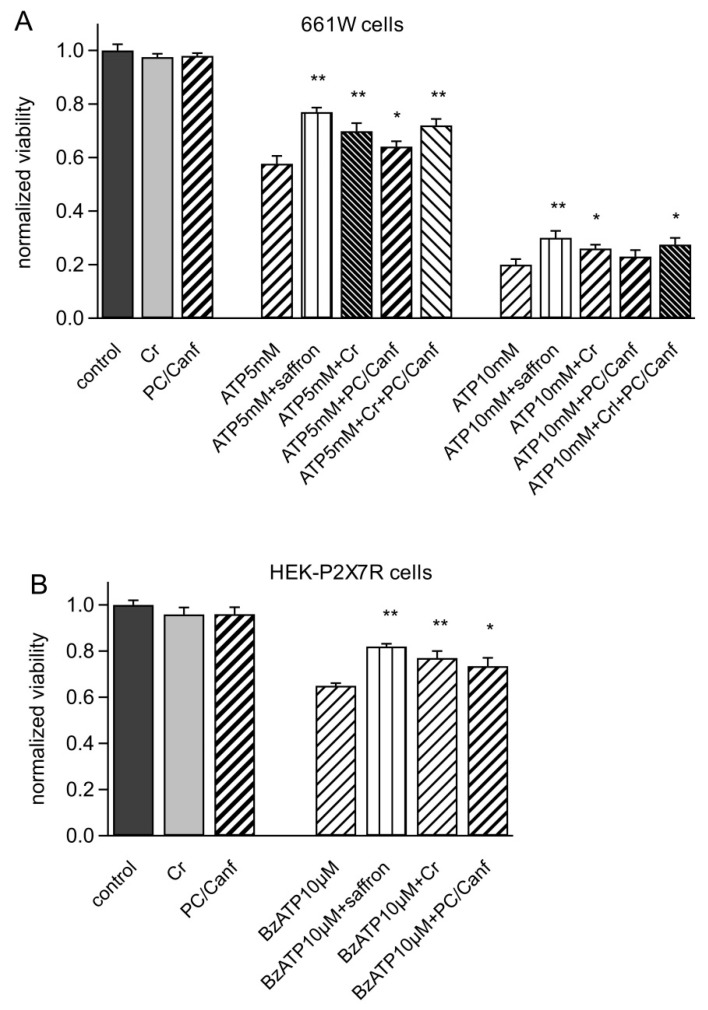
Saffron components increase the viability of 661W and HEK-P2X7R cells. (**A**) Cytotoxic effect in mouse retinal photoreceptor-derived 661W cells induced by application of 5 and 10mM ATP for 24 h and by the co-treatment of ATP saffron, crocins (Cr), picocrocins/kaempferol derivates (PC/Canf) (25 μg/mL). (**B**) Cytotoxicity assay induced on HEK293-P2X7R cells by 10 μM BzATP for 24 h and by the co-treatment of BzATP with 25 μg/mL saffron, Cr, and PC/Canf. Viable cells were counted using an MTT assay (see Materials and Methods) and normalized to control cells. Differences between treatment of ATP and ATP plus saffron and plus the two fractions or BzATP (see Materials and Methods) and BzATP plus saffron and plus the two fractions were significant (* *p* < 0.05, ** *p* < 0.01). Data ± SEM were obtained from triplicates in at least 5 different experiments.

**Figure 4 molecules-25-05618-f004:**
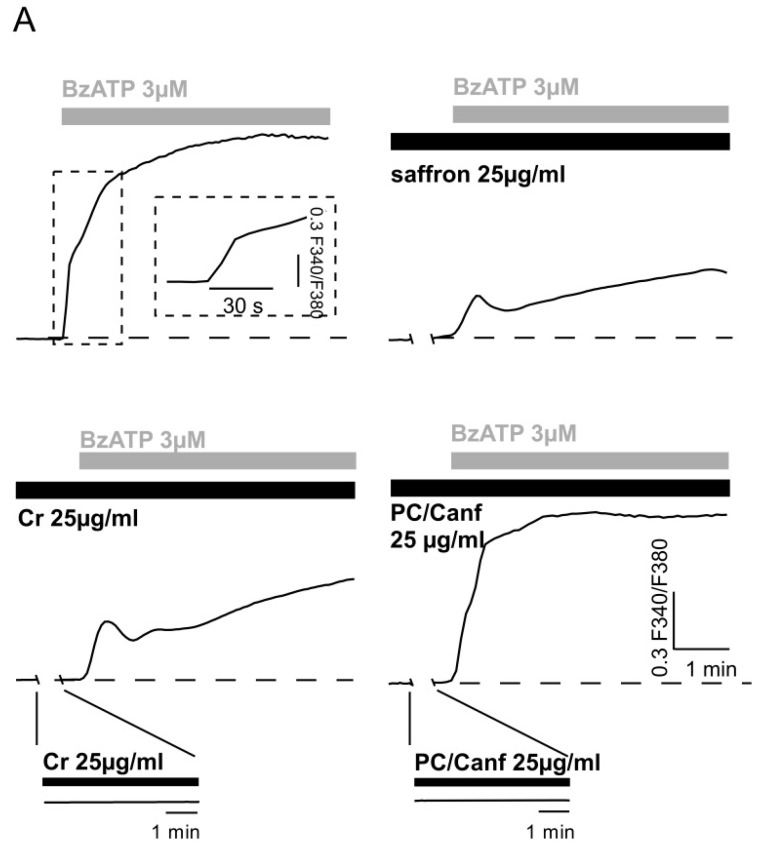
Effect of saffron components on the BzATP-induced [Ca^2+^]_i_ elevation in HEK293-P2X7R cells. (**A**) Representative traces of the fluorescence ratio, indicative of [Ca^2+^]_i_ variation, in response to application of 3 μM BzATP alone (first trace) or to application of BzATP plus 25 μg/mL saffron (second trace), 25 μg/mL crocins (Cr) (third trace), and 25 μg/mL picocrocins/canferols (PC/Canf) (fourth trace). In each experiment, saffron and both fractions were applied 5 min before the application of BzATP. As observed from the insert, the exposure to Cr or PC/Canf alone did not induce any variation of the trace. Horizontal bars indicate the time period of saffron, Cr and PC/Canf (black bars), and BzATP (grey bars) applications. (**B**) The histogram reports the quantitative analysis of [Ca^2+^]_i_ variation elicited by BzATP plus saffron, Cr and PC/Canf. Differences between BzATP and BzATP plus saffron or Cr were significant (** *p* < 0.01), while with PC/Canf no. Data ± SEM were obtained from 44, 61, 44, and 39 cells in the presence of BzATP, plus saffron, plus Cr, and plus PC/Canf, respectively.

**Figure 5 molecules-25-05618-f005:**
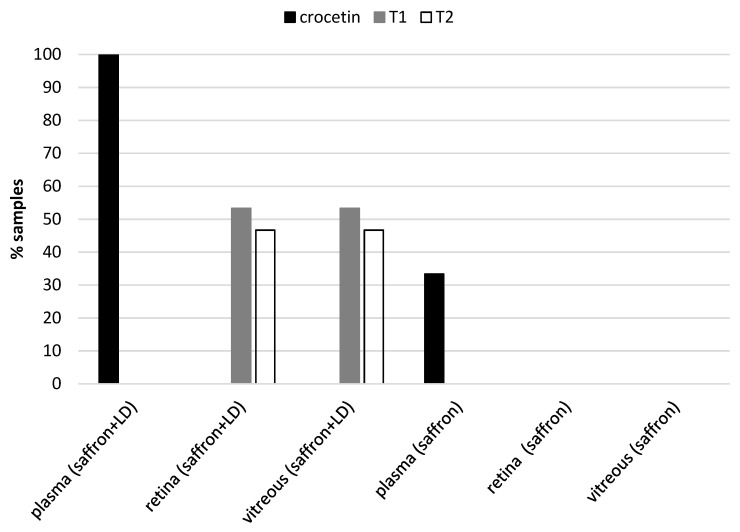
Presence of the main metabolites of saffron: crocetin, trans-crocetin bis (β-d-gentiobiosyl) ester (T1) and trans-crocetin (β-d-gentiobiosyl) (β-d-glucosyl) ester (T2), in the different tissues analyzed. Crocetin is present in plasma samples from animals treated with and without LD, while the two most abundant crocins of saffron are present only in retinal and vitreous samples in about 50% of animals treated and exposed to light damage.

**Table 1 molecules-25-05618-t001:** Statistical analysis of data performed using Student’s *t* test.

	Saffron 1	Saffron 2	Saffron 3	Control	Light Damage
**Saffron 1**		Not significant	*p* < 0.001	*p* < 0.001	Not significant
**Saffron 2**	Not significant		*p* < 0.001	*p* < 0.001	Not significant
**Saffron 3**	*p* < 0.001	*p* < 0.001		*p* < 0.005	*p* < 0.001
**Control**	*p* < 0.001	*p* < 0.001	*p* < 0.005		*p* < 0.001
**Light damage**	Not significant	Not significant	*p* < 0.001	*p* < 0.001

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
