# Peer review of "Saffron: Chemical Composition and Neuroprotective Activity"

_molecules, 2020, doi:10.3390/molecules25235618_

Round 1

Reviewer 1 Report

In their manuscript on chemistry and neuroprotective activity of saffron Maggi et al present a series of different results in relation to the chosen topic including human metabolic, rat in vivo cell toxicity results and cell culture experiments. Each part is for itself interesting, but unfortunately all are lacking many details and remain somehow incomplete. Both, details of methodology and results are critically reduced. Each part would have probably been worth a separate publication with adequate workup of details. The arisen relation to AMD is not sufficiently underlined by scientific data. Therefore conclusions about the significance of these for the pathophysiology or treatment should be re-evaluated.

Further comments:

Materials and Method

The first sentence is incomplete.

Results

Figure 1: It should be “vitreous” instead of “vitreo”.

Page 5 – Lines 208-209: („This suggests….“). This suggestion presumes, that a feedback mechanism exists promoting the transport of these components to the retina. So far no direct evidence is provided for such a hypothetical mechanism. This should critically be discussed.

3.2. Correlation between saffron chemical composition and neuroprotective activity

It is not understandable and surprising why, “it is not possible to appreciate the difference in the contents of crocins by looking at these graphs”. What was the reason for such differences in the chromatograms? Were the analytical methods not identical?

Line 227: The formula should be re-written.

Why did you use albino rats? Phototoxic damage of albino rat is not comparable to pigmented strains. Pigmented strains are probable the only recommendable model of diseased human non-albinotic eyes.

Page 6 – Line 232: It is said, that the experiments were carried out using an animal model of AMD. I cannot recognize an animal model of AMD in the experimental setup. The performed light exposure is not a model of AMD. It is simply a model for phototoxicity.

How long after intensive light exposure were the animals sacrificed?

As far as I can see, the number of animals used in each experiment is not mentioned. Numbers should be added.

The results presented in Fig. 2 and the description in the text should be underlined by statistical evidence. Normally retinal degeneration experiments are examined by electrortinographic methods. Why are these results missing?

Page 7 – Line 259-260: “These results show that…. not all saffron can be used for the treatment of AMD”. First, evidence is missing so far, that any saffron can be used for AMD. Second, as mentioned above, the experiments presented here are not a model of AMD or an analogy of this disease.

Fig. 3: It should be mentioned what kind of staining was used here. I would have preferred sections that show the retinal pigment epithelium as well. Since we see an attenuation of the ONL it is of interest so see the status of the RPE and the choroid. Therefore the sections should be replaced or other histological sections with more informative stainings should be added. The anatomical position of the sections within the retina should be declared. There are many staining techniques that allow an evaluation of the number of nuclei or thickness of layers and much mor information at the same time.

The first sentence of 3.3. is incomplete.

Page 8 – Line 287: Sentence, “Corso et al. showed….” must be corrected.

Fig. 4: I cannot make out the explanations of all abbreviations in the legend of Fig. 4. It would be best to explain the abbreviation in the legend of Fig. 4 because otherwise (even if explanations are somewhere in the text) it is difficult to understand Fig. 4 completely.

In Fig. 4 and 5 abbreviations of the same substances are named in two versions: e.g. BzATP and Bz3 and others as well.

Discussion

I find it unlucky to use the description of “active” and “inactive” saffron. An effect may well be resulting from a passive role of the tested substances.

I find many statements of the discussion problematic on the background that the precise composition and concentrations in the tested fractions are not know. It is suggested to check all statements with regard to these facts.

Page 11 – from line 392 on “The most likely hypotheses…..”: This explanation is not logical: why should all crocin of the body disappear through a damaged blood-retina barrier? This would be very unexpected and unique.

It should be discussed whether the tested concentrations can be achieved by nutrition.

Conclusion

The results do no prove that “saffron treatment would be able to restore correct metabolic pathways”. That is speculation.

How can you prove that “resilience activation requires … positive feedback”?

The sentence, “This complex action is supported…..” is not proven. Why the “entire molecular composition” (something that is so far not completely known!)?

The manuscript needs a thorough workup of style, wording, grammar and linguistic. This should be assisted by a native speaking person or somebody with similar language competence. The submitted text makes the impression of not being checked by the authors after finalization. Some sentences are not complete or are logically inconsistent. Deficits of this kind were mentioned in few cases, but many more can be found in the manuscript.

Author Response

Dear Reviewer, 
The answers to your comments are attached.
In the text of the paper, some of the most important parts modified, have been highlighted.
Yours truly

Reviewer 2 Report

The article has an interesting introduction and findings, however, I would suggest several modifications in order for it to be publishable:

  1. The introduction should include a rationale of the experiments included and the reasons for their selection.
  2. Some kind of scheme or index of the experiments would be desirable, at the beginning of the Materials and Methods section. Also, in L 79, an s should be added to Method.
  3. It is not clear what methods correspond to what experiments, so, this section could be divided into experiments, with their own methods, following the explanation that should be included in the introduction.This information will improve the understanding of the paper.
  4. L 166 says Sprague Dowling, please correct.
  5. L173 on. This paragraph should be clarifyed and L175 soaked should be corrected.
  6. The results section is very confusing. A lot of information is included in it that should either be included in the methods section or the discussion. Please, check other articles to correct this. Only data should be included here, instead there are experimental rationales, hypothesis, judgements of value, data in the literature, etc. All these should be eliminated from this section and allocated somewhere else. 
  7. Also, the results section should be divided in experiments in parallel to the methods section.
  8. L204 It is has not been explained why the researchers chose to perform this measurements and their relation to the rest of the experiments.
  9. Abbreviations are used along the text without prior explanation. See ONL (it is used prior to its explanation), the same happens with AMD. Please check others too.
  10. L 252 on. Results should be described in relation to the LD group.
  11. Statistical data is missing from Fig. 2. Also, in the axis an h should be added to tickness.
  12. Fig.4. 5and 10 doses could be specified in the graph to clarify the graph and the same could be done with the next one. Also, the last columns reads ATP5 when it is supposed to be ATP10
  13. Units are missing from Fig. 5's vertical axis.
  14. The discussion is short, especially considering the abundance of data.
  15. L 394. The idea presented here should be expanded and clarified.
  16. The conclusions should be included in the discussion and only the main findings of the article, with a brief framing, should be included here.

Author Response

(The authors gave the same response as above.)

Round 2

Reviewer 1 Report

In response to the review the authors have performed many corrections and modifications of the manuscript. By this the text was considerably improved.

Author Response

Dear Editor,
only the presence or absence of the crocetin and the two most abundant crocins the different samples is shown in Figure 5. The x-axis indicates only the different types of samples analysed, not a size to which a unit of measurement is associated.
Best regard  
